# Immunogenicity and safety of heterologous boost immunization with PastoCovac Plus against COVID-19 in ChAdOx1-S or BBIBP-CorV primed individuals

Sana Eybpoosh[1], Alireza Biglari[2]*, Rahim Sorouri[3,4], Fatemeh Ashrafian[5], Mona Sadat Larijani[5], Vicente Verez-Bencomo[6], Maria Eugenia Toledo-Romani[7], Carmen Valenzuela Silva[8], Mostafa Salehi-Vaziri[9], Sarah Dahmardeh[10], Delaram Doroud[11], Mohammad Banifazl[12], Ehsan Mostafavi[1], Anahita Bavand[5], Amitis Ramezani[5]*

1 Department of Epidemiology and Biostatistics, Research Centre for Emerging and Reemerging Infectious Diseases, Pasteur Institute of Iran, Tehran, Iran, 2 School of Medicine, Tehran University of Medical Sciences, Tehran, Iran, 3 IPI Directorate, Pasteur Institute of Iran, Tehran, Iran, 4 Department of Microbiology, Faculty of Medicine, Baqiyatallah University of Medical Sciences, Tehran, Iran, 5 Clinical Research Department, Pasteur Institute of Iran, Tehran, Iran, 6 Finlay Vaccine Institute, Havana, Cuba, 7 Pedro Kourí Tropical Medicine Institute, Havana, Cuba, 8 Cybernetics, Mathematics and Physics Institute, Havana, Cuba, 9 COVID-19 National Reference Laboratory, Pasteur Institute of Iran, Tehran, Iran, 10 Vaccination Department, Pasteur Institute of Iran, Tehran, Iran, 11 Quality Control Department, Production and research Complex, Pasteur Institute of Iran, Tehran, Iran, 12 Iranian Society for Support of Patients with Infectious Disease, Tehran, Iran

* biglari63@hotmail.com (AB); amitisramezani@hotmail.com (AR)

**Data Availability Statement:** All data are in the manuscript and/or supporting information files.

## Abstract

### Background

This study aimed at evaluation and comparison of PastoCovac Plus protein-subunit vaccine in parallel with ChAdOx1-S (AstraZeneca) and BBIBP-CorV (Sinopharm) in primarily vaccinated volunteers with two doses of ChAdOx1-S or BBIBP-CorV.

### Materials and methods

194 volunteers enrolled the study who were previously primed with 2 doses of ChAdOx1-S or BBIBP-CorV vaccines. They were divided into two heterologous regimens receiving a third dose of PastoCovac Plus, and two parallel homologous groups receiving the third dose of BBIBP-CorV or ChAdOx1-S. Serum samples were obtained just before and 4 weeks after booster dose. Anti-spike IgG and neutralizing antibodies were quantified and the conventional live-virus neutralization titer, (cVNT50) assay was done against Omicron BA.5 variant. Moreover, the adverse events data were recorded after receiving booster doses.

### Results

ChAdOx1-S/PastoCovac Plus group reached 73.0 units increase in anti-Spike IgG rise compared to the ChAdOx1-S/ ChAdOx1-S (P: 0.016). No significant difference was observed between the two groups regarding neutralizing antibody rise (P: 0.256), indicating equivalency of both booster types. Adjusting for baseline titers, the BBIBP-CorV/PastoCovac Plus

**Funding:** We would acknowledge Pasteur Institute of Iran for financial support of this study (Grant number: 2060 to AR). The funder approved the final protocol, but had no role in study design, data collection, data analysis/interpretation, the decision to publish, or the preparation of the manuscript.

**Competing interests:** The authors have declared that no competing interests exist.

group showed 135.2 units increase ($P$<0.0001) in anti-Spike IgG, and 3.1 ($P$: 0.008) unit increase in mean rise of neutralizing antibodies compared to the homologous group.

Adjustment for COVID-19 history, age, underlying diseases, and baseline antibody titers increased the odds of anti-Spike IgG fourfold rise both in the ChAdOx1-S (OR: 1.9; $P$: 0.199) and BBIBP CorV (OR: 37.3; $P$< 0.0001) heterologous groups compared to their corresponding homologous arms. The odds of neutralizing antibody fourfold rise, after adjustment for the same variables, was 2.4 ($P$: 0.610) for the ChAdOx1-S heterologous group and 5.4 ($P$: 0.286) for the BBIBP CorV heterologous groups compared to their corresponding homologous groups. All the booster types had the potency to neutralize BA.5 variant with no significant difference.

The highest rate of adverse event incidence was recorded for ChAdOx1-S homologous group.

## Conclusions

PastoCovac Plus booster application in primed individuals with BBIBP-CorV or ChAdOx1-S successfully increased specific antibodies' levels without any serious adverse events. This vaccine could be administered in the heterologous regimen to effectively boost humoral immune responses.

### Author summary

PastoCovac Plus is a protein subunit vaccine against COVID-19 which has been assessed highly immunogenic in the conducted clinical trials. According to SARS-CoV-2 ability to escape the immune system, booster doses have been recommended. It has been supposed that a booster of a different type could induce the immune responses stronger than the same priming vaccine series. In the present study, we evaluated PastoCovac Plus protein subunit vaccine as heterologous vaccine regimen in individuals who were primarily immunized with AstraZeneca (Adenovirus-based vaccine) or Sinopharm (inactivated virus-based vaccine) in parallel with the individuals who received a booster of the same priming vaccine (homologous groups). The results showed the great immunogenicity of PastoCovac Plus as a booster dose and increased both anti-Spike IgG and neutralizing antibodies' levels with no serious adverse event. Moreover, the applied booster vaccine successfully neutralized SARS-CoV-2 (Omicron BA.5 variant) through the test. Thus, the findings support mix-and-match strategy regarding COVID-19 vaccination.

## Introduction

COVID-19 as a global health concern, has resulted in several vaccine strategies with the hope of elimination, though it is still developing through new variants [1–4]. The administration of the approved vaccines can induce herd immunity and lead to achieve protection against the infection acquisition or severe form of the disease, however, many studies proved that re-infection occurs in previously infected individuals [5,6].

What is more, cohort studies regarding COVID-19, have claimed that specific antibodies wane over time, a fact which has been recorded for all vaccine platforms [7–10]. Furthermore, new SARS-CoV-2 variants with the potency of immune system evasion and increased risk of

infectivity, have demanded a booster dose to enhance the immune system responses [11,12]. As the investigated trials grew up, the heterologous regimens showed considerable efficiency to boost the waned immune responses and came to attention due to the ease of supply especially for regions with limited source of vaccines beside the immunogenicity potency [13].

In other words, the urgent need of COVID-19 vaccine development has led to a parallel shift to heterologous administration to save time. Subsequently, the availability of different vaccine candidates has brought the opportunity of heterologous prime-boost vaccination strategies to elicit stronger and broader humoral and cellular immune responses [14,15].

For instance, a heterologous booster vaccine, six months after the second dose of Corona-Vac enhanced the protection against COVID-19 [16]. The primed subjects with AZD1222, generated more neutralizing antibodies after BNT16b2 booster shot in comparison with the homologous booster recipients [17].

Protein subunit vaccines have a successful safe and effective profile among the vaccine platforms [18–20]. However, the effectiveness of mix-and-match strategy need to be more investigated to come up with the best heterologous prime-boost schedules, preferably with the lowest rate of adverse events [21].

PastoCovac (SOBERANA 02) as the primary vaccine dose and PastoCovac Plus (SOBERANA Plus) as the booster shot, are protein subunit vaccines based on SARS-CoV-2 RBD, [Arg319-Phe541-(His)6]. PastoCovac Plus contains 50 µg of dimerized RBD (d-RBD) applying aluminium hydroxide as adjuvant. These vaccines were primarily developed at the Finlay Vaccine Institute of Cuba, and then have been co-developed and manufactured in Pasteur Institute of Iran after successful technology transfer [22].

In this study we evaluated the safety and immunogenicity of PastoCovac Plus as a booster dose in parallel with ChAdOx1-S (AstraZeneca) and BBIBP-CorV (Sinopharm) in vaccinated individuals with two doses of ChAdOx1-S or BBIBP-CorV. Furthermore, the booster immunogenicity and adverse events via heterologous or homologous regimens were compared.

## Materials and methods

### Ethics statement

This study was approved by the ethical board of the Pasteur Institute of Iran (Reference number: IR.PII.REC.1400.076). All included participants were provided with informed consent for scientific analyses and a written consent was obtained from the participants. All the methods were performed according to the Declaration of Helsinki and with the relevant guidelines and regulations.

### Study participants

This longitudinal study was conducted on 194 volunteers (>18 years) of Pasteur Institute of Iran in a follow-up schedule from January to the end of June, 2022 who met the agreement on the criteria of the study. Fig 1 simply presents the study design.

The pregnant and breast-feeding women, immunocompromised patients, participants with an uncontrolled underlying disease were not included in the study according to the physician's screen.

### Vaccine group design

The participants were divided into groups as:

I) The heterologous vaccine groups; a) The ChAdOx1-S/PastoCovac Plus prime-boost group in which participants received two doses of ChAdOx1-S [AstraZeneca, Oxford (12–16

## Immunogenicity and safety of heterologous boost immunization with PastoCovac Plus® against COVID-19 in ChAdOx1-S or BBIBP-CorV primed individuals

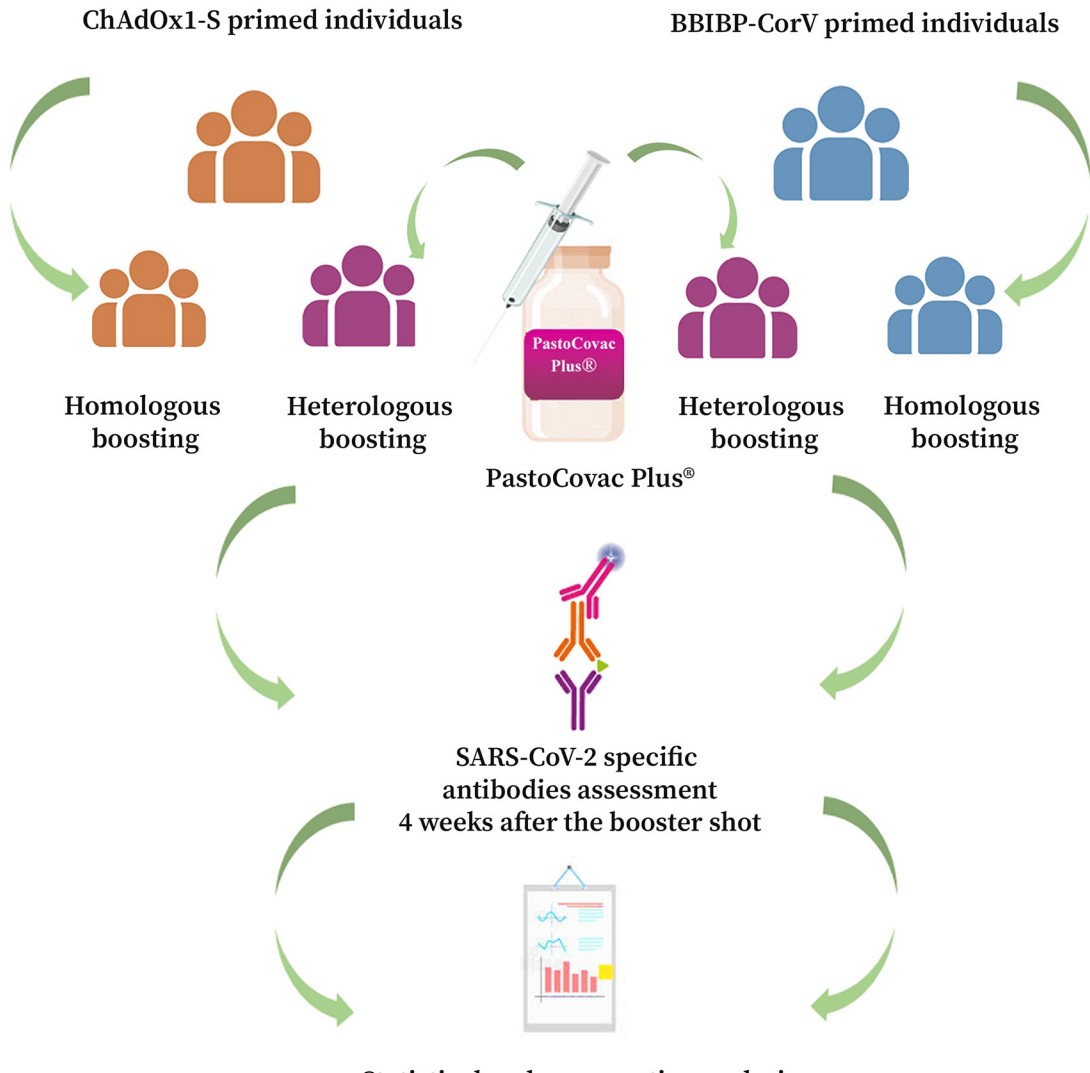

**Fig 1. The study arms and follow-up process in the study.**

weeks intervals], and a booster dose of PastoCovac Plus b) The BBIBP /PastoCovac Plus prime-boost group; in which the participants received two doses of BBIBP vaccine [Sinopharm, Beijing CNBG (4–5 weeks intervals)] and a booster shot of PastoCovac Plus.

II) The homologous vaccine groups; a) The ChAdOx1-S prime-boost group who got three doses of ChAdOx1-S, b) BBIBP prime-boost group who got three doses of BBIBP vaccine.

## Immunogenicity evaluation

Blood samples were collected upon the admission for the booster shot and also on day 28±5 after it. Upon serum isolation, assessment of the generated antibodies was investigated with titers of Anti-SARS-CoV-2 Quantivac ELISA (IgG) (Euroimmun, Lübeck, Germany) and SARS-CoV-2 Neutralizing Ab (Pishtazteb, Iran).

## Virus neutralizing test

Conventional neutralizing antibody titers (cVNT50) were evaluated on randomly selected samples [23]. Briefly, the samples were inactivated at 56˚C for 30 mins. Vero cells were seeded in DMEM containing 10% FBS. Serial dilution of sera samples was prepared. Then 50 μl of each serum was mixed with 50 μl of 100 $TCID_{50}$ of SARS-CoV-2 Omicron BA.5 variant and incubated at room temperature for an hour. Next, the prepared mixture was added into the wells containing monolayers of Vero cells for 60 min incubation at 37˚C. The control wells were as a well without the mixture of serum and virus, one well without any serums, and one well without any cells. After the incubation time and supernatant removal, the cells were washed with DMEM. After 72 hrs of maintaining in DMEM at 37˚C, the CPE was assessed applying an inverted microscope and the titer of neutralizing antibodies was evaluated in accordance to the highest serum dilution in which the virus was neutralized in 50% of the wells.

## Adverse event assessment

All the local and systemic adverse events were documented in the appropriate questionnaire through a phone call one week and an interview four weeks post the booster dose.

## Statistical analysis

Fourfold rise of anti-SARS-CoV-2-Spike IgG and neutralizing antibodies was defined as fourfold rise in antibody titers compared to the baseline (the day of booster injection). Antibody rise was calculated by subtracting antibody titers on day 28±5 from the baseline. Fold rise of the antibodies was calculated by dividing antibody titers on day 28±5 to the baseline. The Geometric Mean Titer (GMT) of anti-SARS-CoV-2-Spike IgG and neutralizing antibodies, titer rise, and fold rise were calculated in the immunogenicity analysis. 95% confidence interval (CI) for GMT ratio were calculated using nonparametric percentile bootstrap method with 1000 repetitions.

Titer levels were compared between each heterologous group and its corresponding homologues group after adjustment for variations in pre-boost titers, as well as age, COVID-19 infection history, and underlying medical diseases, using Quintile regression analysis. Titer levels before and 4-weeks after booster injection was compared within each group using Wilcoxon test. The frequency of participants with fourfold rise was compared between the groups using Pearson Chi Square test. Fisher exact test was used as an alternative test when more than 5% of cells had frequency lower than 5. Logistic regression analysis was done in order to investigate crude and adjusted effect of booster type on antibody fourfold rise rates.

The frequency of the reported adverse events after the booster dose was compared between homologous and heterologous groups using the Chi-square or Fisher's exact tests.

# Results

## Demographic characteristics

A total of 194 volunteers including 90 (46.4%) men and 104 (53.6%) women with the mean age of 45±13.6 (19–89) years old were investigated. 38 individuals had at least one underlying

**Table 1. Demographic and clinical characteristics of volunteers in the homologous and heterologous ChAdOx1-S and BBIBP-CorV vaccine groups.**

| | Total n (%) | ChAdOx1-S (AstraZeneca) | | | BBIBP-CorV (Sinopharm) | | |
| --- | --- | --- | --- | --- | --- | --- | --- |
| | | Homologous ChAdOx1-S n (%) | Heterologous ChAdOx1-S /PastoCovac Plus n (%) | p value | Homologous BBIBP-CorV N (%) | Heterologous BBIBP CorV/ PastoCovac Plus N (%) | p value |
| **Overall volunteers** | 194 (100) | 27 (13.9) | 67 (34.5) | | 50 (25.8) | 50 (25.8) | |
| **Age** | | | | 0.975$^\S$ | | | 0.263$^\S$ |
| Mean (SD) | 45.0 (13.6) | 43.0 (12.4) | 43.1 (11.2) | | 45.1 (15.3) | 48.6 (15.2) | |
| Min, Max | 19.89 | 26.73 | 25. 71 | | 19. 89 | 24. 81 | |
| **Sex** | | | | 0.712* | | | 0.689* |
| Female | 104 (53.6) | 12 (44.4) | 40 (59.7) | | 25 (50.0) | 27 (54.0) | |
| Male | 90 (46.4) | 15 (55.6) | 27(40.3) | | 25 (50.0) | 23 (46.0) | |
| **BMI** | | | | 0.434$^\S$ | | | 0.738$^\S$ |
| Mean (SD) | 25.8 (3.9) | 26.1 (4.2) | 25.4 (3.9) | | 26.1 (4.2) | 25.8 (3.4) | |
| Min, Max | 18.2, 37.2 | 18.9, 35.5 | 18.9, 36.3 | | 18.2, 37.2 | 19.8, 35.6 | |
| < 25 | 90 (46.4) | 10 (37.0) | 37 (55.2) | | 23 (46.0) | 20 (40.0) | |
| ≥ 25 | 104 (53.6) | 17 (63.0) | 30 (44.8) | | 27 (54.0) | 30 (60.0) | |
| **COVID-19 History** | | | | | | | |
| Before vaccination | 68 (35.1) | 12 (44.4) | 27 (40.3) | 0.712* | 12 (24.0) | 17 (34.0) | 0.271* |
| Between dose 1 & 2 | 15 (7.7) | 4 (14.8) | 6 (9.0) | 0.404** | 0 (0) | 5 (10.0) | 0.056** |
| Between dose 2 & 3 | 2 (1.0) | 1 (3.7) | 0 (0) | 0.287** | 0 (0) | 1 (2.0) | 0.500** |
| Up to 2 weeks after Booster | 4 (2.1) | 2 (7.4) | 0 (0) | 0.080** | 0 (0) | 2 (4) | 0.495* |
| Any history of COVID-19 | 74 (38.1) | 14 (51.6) | 27 (40.3) | 0.307* | 12 (24.0) | 21 (42.0) | 0.056* |
| **Re-infection$^¥$** | 20 (10.3) | 4 (14.8) | 11 (16.4) | 0.848** | 0 (0) | 5 (10.0) | 0.056* |
| **Vaccine breakthrough** | 26 (13.4) | 3 (11.1) | 3 (19.4) | 0.333** | 4 (8.0) | 6 (12.0) | 0.741* |
| **Underlying disease** | | | | | | | |
| No | 156 (80.4) | 22 (81.5) | 61 (91.0) | 0.192* | 34 (68.0) | 39 (78.0) | 0.260* |
| Yes | 38 (19.6) | 5 (18.5) | 6 (9.0) | | 16 (32.0) | 11 (22.0) | |

* Pearson Chi-Square

** Fisher's Exact Test

§ Mann-Whitney U. CI: confidence interval.

As some of the participants had reinfection, the number under the "any history of COVID-19" row, does not equal the sum of above rows.

$^¥$ Re-infection is defined as having COVID-19 infection ≥ 2 times from the onset of the pandemic till the end of the follow-up period.

Bold P values are indicated statistically significant.

disease. The most prevalent underlying disease among the participants was hypertension (6.7%). There was no significant difference between the studied groups in terms of age, gender or underlying disease in the studied groups. The details of demographic data are presented in Table 1.

The population of each group was as ChAdOx1-S/ PastoCovac Plus prime-boost: n = 67, ChAdOx1-S prime-boost: n = 27, BBIBP-CorV/PastoCovac Plus prime-boost: n = 50 and BBIBP-CorV prime-boost: n = 50.

The evaluation of COVID-19 history of the participants revealed that 38.1% developed the infection in different window times. In addition, 26 individuals got infected two-weeks post-the booster shot during the follow up which represents the vaccine breakthrough. Finally, 10.3% experienced re-infection during the study (Table 1).

## Antibody assessment

The geometric mean titer (GMT) of anti SARS-CoV-2-Spike IgG and neutralizing antibodies were evaluated before and 4 weeks after booster injection (Tables 2 and 3). Anti-Spike IgG rise was seen in all booster recipients whether in homologous or heterologous pattern (Fig 2). Nevertheless, neutralizing antibody presented differently between the groups as the following categories describe (Fig 3).

## Adenovirus-based booster (ChAdOx1-S) vs. protein subunit booster (PastoCovac Plus)

Anti-Spike IgG in ChAdOx1-S/ PastoCovac Plus arm increased from 64.1 (95% CI: 46.9–87.5) to 310.5 (95% CI: 232.9–413.8) whereas in ChAdOx1-S homologous group showed a mean of 72.5 (95% CI: 55.0–95.6) units increase. After adjustment for baseline of anti-Spike IgG, age, COVID-19 and underlying diseases, anti-Spike IgG rise was 70.3 units higher in ChAdOx1-S/ PastoCovac Plus group than the homologous group (P: 0.008; Table 2). Nevertheless, there was no significant difference in neutralizing antibody rise between the two groups after adjustment for the same variables (Table 3). In addition, fourfold rise of anti-Spike IgG and neutralizing antibody was not significantly different between the two groups (Tables 2 and 3).

**Table 2. Anti-Spike IgG geometric mean titer (GMT), mean rise, fold rise, and fourfold rise rate between ChAdOx1-S and BBIBP-CorV homologous and heterologous groups.**

| ChAdOx1-S | Homologous (n = 27) | Heterologous (n = 67) | Crude | | Adjusted$^\alpha$ | |
|---|---|---|---|---|---|---|
| | | | Beta$^£$ | P value | Beta$^£$ | P value |
| Before GMT (95% CI) | 36.4 (26.0, 50.9) | 64.1 (46.9, 87.5) | 19.1 | 0.247 | - | - |
| After GMT (95% CI) | 123.5 (98.7, 154.4) | 310.5 (232.9, 413.8) | 81.4 | **0.038** | 70.3 | **0.008** |
| Rise GMT (95% CI) | 72.5 (55.0, 95.6) | 198.8 (146.5, 269.6) | 86.8 | **0.006** | 70.3 | **0.008** |
| Fold Rise GMT (95% CI) | 3.4 (2.6, 4.4) | 4.8 (3.8, 6.1) | 1.1 | 0.299 | 0.7 | 0.487 |
| Fourfold rise, % | 11 (40.7) | 36 (53.7) | 1.7$^§$ | 0.257 | 1.9$^§$ | 0.199 |
| BBIBP-CorV | Homologous (n = 50) | Heterologous (n = 50) | | | | |
| Before GMT (95% CI) | 30.4 (20.1, 45.8) | 14.8 (9.5, 22.9) | -16.0 | 0.059 | - | - |
| After GMT (95% CI) | 92.9 (58.4, 147.8) | 195.5 (147.5, 259.2) | 120.4 | **0.003** | 134.6 | **<0.0001** |
| Rise GMT (95% CI) | 32.8 (18.0, 59.8) | 167.5 (126.8, 221.4) | 134.9 | **<0.0001** | 134.6 | **<0.0001** |
| Fold Rise GMT (95% CI) | 3.1 (2.2, 4.4) | 13.2 (9.6, 18.3) | 9.9 | **<0.0001** | 8.5 | **0.008** |
| Fourfold rise, % | 13 (26.0) | 46 (92.0) | 32.7$^§$ | **<0.0001** | 37.3$^§$ | **<0.0001** |

* Pearson Chi-Square

** Fisher's Exact Test; CI: confidence interval.

$^§$ The values indicate the ORs generated from logistic regression analysis.

$^£$ Beta coefficients and P values are generated from quantile regression method.

$^\alpha$ Adjusted for age, COVID-19, underlying diseases, and baseline Ab titer. Bold P value are indicated statistically significant.

**Table 3. Neutralizing antibody geometric mean titer (GMT), mean rise, fold rise, and fourfold rise rate between ChAdOx1-S and BBIBP-CorV homologous and heterologous groups.**

| ChAdOx1-S | Homologous (n = 27) | Heterologous (n = 67) | Crude | | Adjusted | |
|---|---|---|---|---|---|---|
| | | | Beta$^£$ | *P* value | Beta$^£$ | *P* value |
| **Before** GMT (95% CI) | 11.0 (6.3, 19.4) | 10.4 (7.3, 14.9) | -5.9 | 0.373 | - | - |
| **After** GMT (95% CI) | 30.4 (24.7, 37.5) | 32.4 (30.4, 34.5) | 0.2 | 0.815 | 1.0 | 0.427 |
| **Rise** GMT (95% CI) | 7.4 (4.1, 13.1) | 7.9 (5.4, 11.4) | 4.0 | 0.440 | 1.0 | 0.427 |
| **Fold Rise** GMT (95% CI) | 2.8 (1.7, 4.5) | 3.1 (2.2, 4.4) | 0.4 | 0.572 | 0.9 | 0.966 |
| **Fourfold Rise**, % | 7 (25.9) | 18 (26.9) | 1.1$^§$ | 0.926 | 2.4$^§$ | 0.610 |
| **BBIBP-CorV** | **Homologous (n = 50)** | **Heterologous (n = 50)** | | | | |
| **Before** GMT (95% CI) | 20.3 (16.5, 24.9) | 2.4 (1.3, 4.4) | -23.9 | **<0.0001** | - | - |
| **After** GMT (95% CI) | 28.9 (26.3, 31.7) | 31.2 (26.9, 36.1) | 1.1 | 0.334 | 3.3 | **0.008** |
| **Rise** GMT (95% CI) | 1.2 (0.6, 2.3) | 15.7 (11.2, 22.2) | 25.6 | **<0.0001** | 3.3 | **0.008** |
| **Fold Rise** GMT (95% CI) | 1.4 (1.2, 1.7) | 12.8 (7.1, 23.2) | 17.7 | **0.005** | 2.1 | 0.557 |
| **Fourfold Rise**, % | 2 (4.0) | 28 (56.0) | 30.6$^§$ | **<0.0001** | 5.4$^§$ | 0.286 |

\* Pearson Chi-Square

\*\* Fisher's Exact Test; CI: confidence interval.

$^§$ The values indicate the ORs generated from logistic regression analysis.

$^£$ Beta coefficients and P values are generated from quantile regression method.

$^α$ Adjusted for age, COVID-19, underlying diseases, and baseline Ab titer. Bold *P* value are indicated statistically significant.

## Inactivated virus-based booster (BBIBP CorV) vs. protein subunit booster (PastoCovac Plus)

The anti-spike IgG titer in the BBIBP CorV /PastoCovac Plus prime-boost group increased from 14.8 (95% CI: 9.5–22.9) to 195.5 (95% CI: 147.5–259.2) whereas in the BBIBP-CorV homologous group, it showed a mean of 32.8 (95% CI: 18.0–59.8) units increase. After adjustment for baseline anti-Spike IgG titers, age, COVID-19 and underlying diseases, the anti-Spike IgG rise was 134.6 units higher in BBIBP CorV /PastoCovac Plus group than the homologous group (*P*<0.0001; Table 2). Anti-Spike IgG fourfold rise rate in the BBIBP CorV heterologous group was 46 (92%) whereas 13 (26%) in the homologous regimen (*P*<0.0001, Table 2). Neutralizing antibody mean rise and fold rise in the heterologous group were 3.3 and 2.1 units higher in comparison to the homologous group after adjustment for the same variables (*P*: 0.008 and *P*: 0.557, respectively; Table 3).

## The impact of COVID-19 history on specific antibodies among the study groups

COVID-19 infection history did not significantly affect anti-Spike IgG fourfold rise or mean rise in any study arms. Nevertheless, it had a significant impact on neutralizing antibody fourfold rise and mean rise among heterologous ChAdOx1-S (*P*<0.0001 and *P*: 0.0002, respectively; S2 Table and Figs 2 and 3). Thus, upon adjustment for COVID-19 infection history, the odds of neutralizing antibody fourfold rise after heterologous boosting decreased from 30.5 to 2.5 times and the association was no longer statistically significant in comparison with the homologous booster dose, (Table 4, Models 1 and 3).

## The association of age with specific antibodies among the study groups

In order to assess the effect of age on antibody trend after the booster shot, the participants of each group were divided as ≥50 and <50 years old. The results showed that, there was no

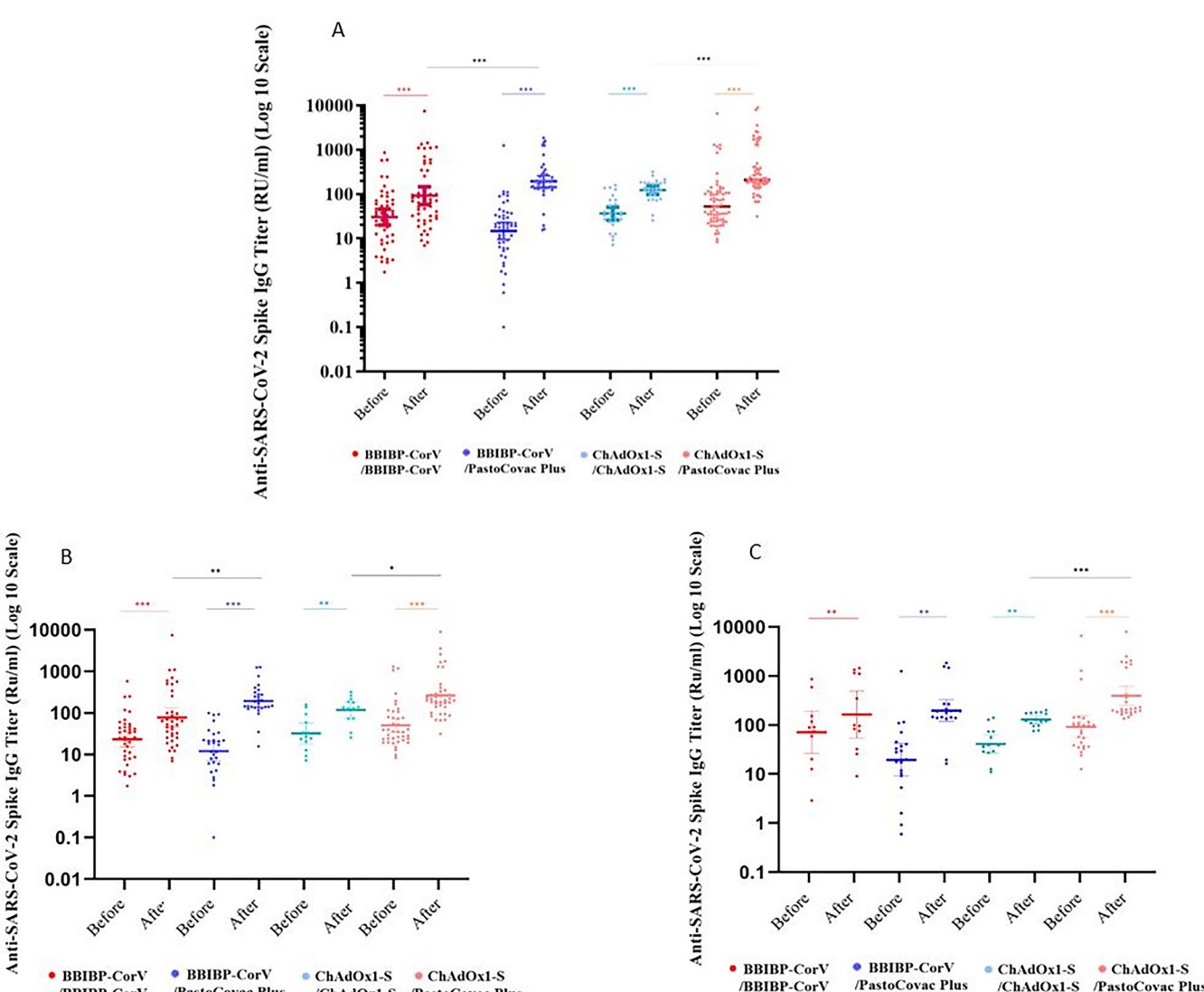

**Fig 2. Scatter plot of anti-Spike IgG antibody responses before and four weeks after the booster dose in four study groups. A) General; B) antibody status in individuals without COVID-19 infection history; and C) antibody status in individuals without COVID-19 infection history.** Geometric mean titer on log 10 scale and 95% confidence intervals are shown. Within-group differences (before and after the booster injection) are estimated using the Wilcoxon match pair test. Differences in antibody titer after the booster injection between homologues and heterologous groups are estimated using Mann-Whitney U test. Significant differences are indicated within the figures; *** indicates *P* values that are significant at <0.0001 levels, ** indicates *P* values less than 0.01, and * indicates *P* values less than 0.05 levels.

significant difference in terms of anti-Spike IgG mean rise and fourfold rise between individuals ≥50 or <50 years of ChAdOx1-S and BBIBP CorV through homologous or heterologous regimens (S1 and S2 Tables).

However, neutralizing antibody mean rise and fourfold rise rate were significantly higher in the heterologous ChAdOx1-S /PastoCovac Plus <50 years individuals (*P* = 0.0063, *P* = 0.028; respectively).

In contrary to ChAdOx1-S primed individuals, age did not affect BBIBP-CorV primed group, either through homologous or heterologous regimens (S1 and S2 Tables).

Fourfold rise of anti-Spike IgG and neutralizing antibody was 1.9 and 2.4 times greater in the ChAdOx1-S/PastoCovac Plus heterologous group in comparison to the ChAdOx1-S

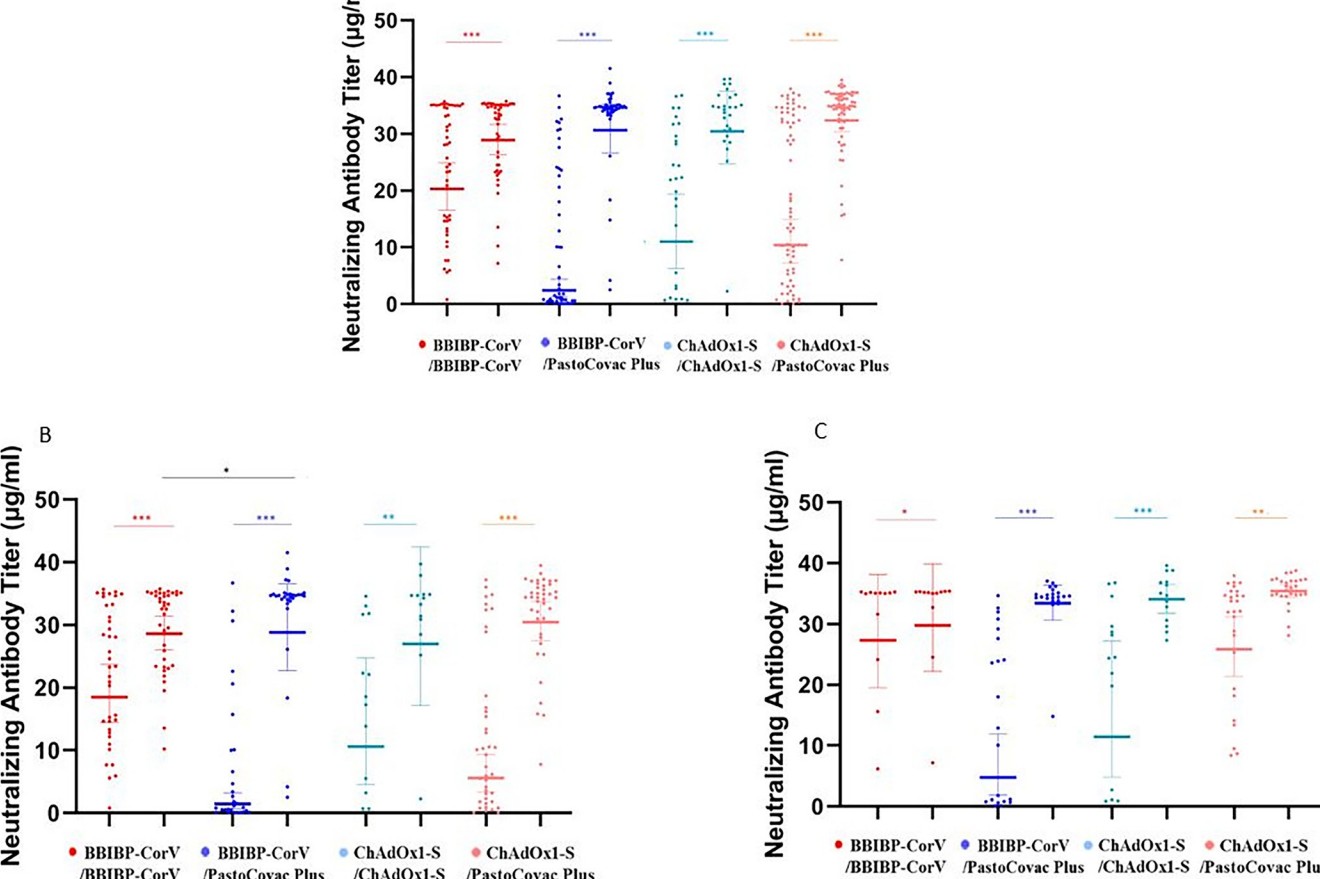

**Fig 3. Scatter plot of neutralizing antibody responses before and four weeks after the booster dose in four study groups. A) General; B) antibody status in individuals without COVID-19 infection history; and C) antibody status in individuals without COVID-19 infection history.** Geometric mean titer on log 10 scale and 95% confidence intervals are shown. Within-group differences (before and after the booster injection) are estimated using the Wilcoxon match pair test. Differences in antibody titer after the booster injection between homologues and heterologous groups are estimated using Mann-Whitney U test. Significant differences are indicated within the figures; *** indicates $P$ values that are significant at <0.0001 levels, ** indicates $P$ values less than 0.01, and * indicates $P$ values less than 0.05 levels.

homologous group adjustment for COVID-19 history, age, comorbidity, and baseline antibody titers. The associations, however, were not statistically significant (Table 4). This finding indicates equivalency in immune response between both types of boosters in recipients of ChAdOx1-S.

The odds of anti-Spike IgG fourfold rise in BBIBP CorV/PastoCovac Plus heterologous group was 37.3 times higher than the BBIBP CorV homologous group after adjustment for COVID-19 history, age, comorbidity, and baseline antibody titers ($P$ <0.0001). In parallel, neutralizing antibody fourfold rise rate was 5.4 folds higher in BBIBP CorV/PastoCovac Plus recipients than the homologous group after adjustment for the above variables ($P$: 0.286; Table 4).

## Neutralization assay

The virus neutralization potency of the induced antibodies was assessed through cVNT50 test on randomly selected sera samples against Omicron BA.5 variant (Fig 4). The results showed that all the booster types neutralized Omicron BA.5 variant though there was no significant difference.

**Table 4. Crude and adjusted effect of homologous vs. heterologous boosters on anti-Spike IgG and neutralizing Ab fourfold rise in primed recipients with ChAdOx1-S and BBIBP CorV.**

| | Anti-Spike IgG Fourfold rise (Yes/No) | | | Neutralizing Ab Fourfold rise (Yes/No) | | |
|---|---|---|---|---|---|---|
| PastoCovac Plus vs. ChAdOx1-S | OR | 95% CI | *P*- value | OR | 95% CI | *P*- value |
| **Crude** | | | | | | |
| PastoCovac Plus | 1.7 | 0.7, 4.2 | 0.257 | 1.0 | 0.4, 2.9 | 0.926 |
| ChAdOx1-S (Ref) | - | - | | | | |
| **Model 1** | | | | | | |
| PastoCovac Plus | 2.2 | 0.9, 5.4 | 0.106 | 2.0 | 0.1, 36.2 | 0.640 |
| ChAdOx1-S (Ref) | - | - | | - | - | |
| Baseline Ab Titer$^\S$ | 1.0 | 0.9, 1.1 | 0.087 | 0.5 | 0.3, 0.8 | 0.001 |
| **Model 2** | | | | | | |
| PastoCovac Plus | 2.1 | 0.8, 5.4 | 0.108 | 1.8 | 0.1, 37.5 | 0.698 |
| ChAdOx1-S (Ref) | - | - | | - | - | |
| Age (Years) | 1.0 | 0.9, 1.1 | 0.802 | 1.0 | 0.8, 1.1 | 0.807 |
| Baseline Ab Titer$^\S$ | 1.0 | 0.9, 1.0 | 0.094 | 0.5 | 0.3, 0.8 | 0.001 |
| **Model 3** | | | | | | |
| PastoCovac Plus | 2.0 | 0.8, 5.1 | 0.149 | 1.7 | 0.1, 35.1 | 0.733 |
| ChAdOx1-S (Ref) | - | - | | - | - | |
| COVID-19 (Yes/ No) | 0.6 | 0.2, 1.2 | 0.107 | 0.6 | 0.1, 11.0 | 0.721 |
| Baseline Ab Titer$^\S$ | 1.0 | 0.9, 1.1 | 0.080 | 0.5 | 0.3, 0.8 | 0.001 |
| **Model 4** | | | | | | |
| PastoCovac Plus | 2.0 | 0.8, 5.2 | 0.150 | 3.0 | 0.1, 69.0 | 0.486 |
| ChAdOx1-S (Ref) | - | - | | - | - | |
| Underlying disease (Yes/ No) | 0.3 | 0.1, 1.3 | 0.117 | 0.2 | 0.1, 2.5 | 0.179 |
| Baseline Ab Titer$^\S$ | 1.0 | 0.9, 1.0 | 0.070 | 0.5 | 0.3, 0.8 | 0.002 |
| **Model 5** | | | | | | |
| PastoCovac Plus | 1.9 | 0.7, 4.9 | 0.199 | 2.4 | 0.1, 67.0 | 0.610 |
| ChAdOx1-S (Ref) | - | - | | - | - | |
| Age (Years) | 1.0 | 0.9, 1.0 | 0.876 | 1.0 | 0.8, 1.1 | 0.699 |
| COVID-19 (Yes/ No) | 0.5 | 0.2, 1.2 | 0.125 | 0.4 | 0.1, 10.1 | 0.579 |
| Underlying disease (Yes/No) | 0.3 | 0.1, 1.4 | 0.135 | 0.1 | 0.0, 2.6 | 0.172 |
| Baseline Ab Titer$^\S$ | 1.0 | 0.9, 1.0 | 0.076 | 0.5 | 0.3, 0.8 | 0.003 |
| **PastoCovac Plus vs. BBIBP CorV** | | | | | | |
| **Crude** | | | | | | |
| PastoCovac Plus | 32.7 | 9.8, 108.8 | **<0.0001** | 30.5 | 6.7, 139.8 | **<0.0001** |
| BBIBP CorV (Ref) | - | - | | - | - | |
| **Model 1** | | | | | | |
| PastoCovac Plus | 37.0 | 9.5, 143.0 | **<0.0001** | 2.5 | 0.2, 39.8 | 0.517 |
| BBIBP CorV (Ref) | - | - | | - | - | |
| Baseline Ab Titer$^\S$ | 1.0 | 0.9, 1.0 | 0.110 | 0.5 | 0.3, 0.8 | **0.002** |
| **Model 2** | | | | | | |
| PastoCovac Plus | 38.1 | 9.6, 150.7 | **<0.0001** | 3.7 | 0.2, 66.7 | 0.383 |
| BBIBP CorV (Ref) | - | - | | - | - | |
| Age (Years) | 1.0 | 0.9, 1.1 | 0.161 | 0.9 | 0.8, 1.0 | 0.253 |
| Baseline Ab Titer$^\S$ | 1.0 | 0.9, 1.1 | 0.118 | 0.5 | 0.3, 0.8 | **0.002** |
| **Model 3** | | | | | | |
| PastoCovac Plus | 36.5 | 9.1, 146.0 | **<0.0001** | 2.5 | 0.2, 39.3 | 0.515 |
| BBIBP CorV (Ref) | - | - | | - | - | |

*(Continued)*

**Table 4.** (Continued)

| PastoCovac Plus vs. ChAdOx1-S | Anti-Spike IgG Fourfold rise (Yes/No) | | | Neutralizing Ab Fourfold rise (Yes/No) | | |
|---|---|---|---|---|---|---|
| | OR | 95% CI | P- value | OR | 95% CI | P- value |
| COVID-19 (Yes/ No) | 1.1 | 0.3, 4.4 | 0.936 | 1.1 | 0.1, 20.5 | 0.939 |
| Baseline Ab Titer$ | 1.0 | 0.9, 1.0 | 0.119 | 0.5 | 0.3, 0.8 | **0.002** |
| **Model 4** | | | | | | |
| PastoCovac Plus | 35.9 | 9.2, 140.2 | **<0.0001** | 3.7 | 0.2, 69.7 | 0.382 |
| BBIBP CorV (Ref) | - | - | | - | - | |
| Underlying disease (Yes/ No) | 0.5 | 0.2, 1.8 | 0.280 | 0.2 | 0.0, 3.5 | 0.249 |
| Baseline Ab Titer$ | 1.0 | 0.9, 1.0 | 0.113 | 0.5 | 0.3, 0.8 | **0.003** |
| **Model 5** | | | | | | |
| PastoCovac Plus | 37.3 | 8.8, 157.8 | **<0.0001** | 5.4 | 0.2, 119.9 | 0.286 |
| BBIBP CorV (Ref) | - | - | | - | - | |
| Age (Years) | 1.0 | 0.9, 1.1 | 0.108 | 1.0 | 0.9, 1.1 | 0.303 |
| COVID-19 (Yes/ No) | 0.9 | 0.2, 4.0 | 0.960 | 0.7 | 0.1, 14.4 | 0.786 |
| Underlying disease (Yes/No) | 0.4 | 0.1, 1.6 | 0.184 | 0.2 | 0.1, 4.8 | 0.304 |
| Baseline Ab Titer$ | 0.9 | 0.8, 1.0 | 0.136 | 0.5 | 0.3, 0.8 | **0.003** |

Bold P values are statistically significant. $ In each group, baseline antibody refers to the antibody titer measured before receiving the 3rd dose. In models assessing the effect of different variables on Anti-Spike IgG Fourfold rise, baseline Ab Titer refers to the baseline Anti-Spike IgG. In models assessing the effect of different variables on Neutralizing Ab Fourfold rise, baseline Ab Titer refers to the baseline Neutralizing Ab.

### Evaluation of adverse events post the booster shots

The adverse events were recorded to be compared between the homologous and the heterologous regimens following the booster injection of which the most frequent ones were local pain, fatigue or weakness.

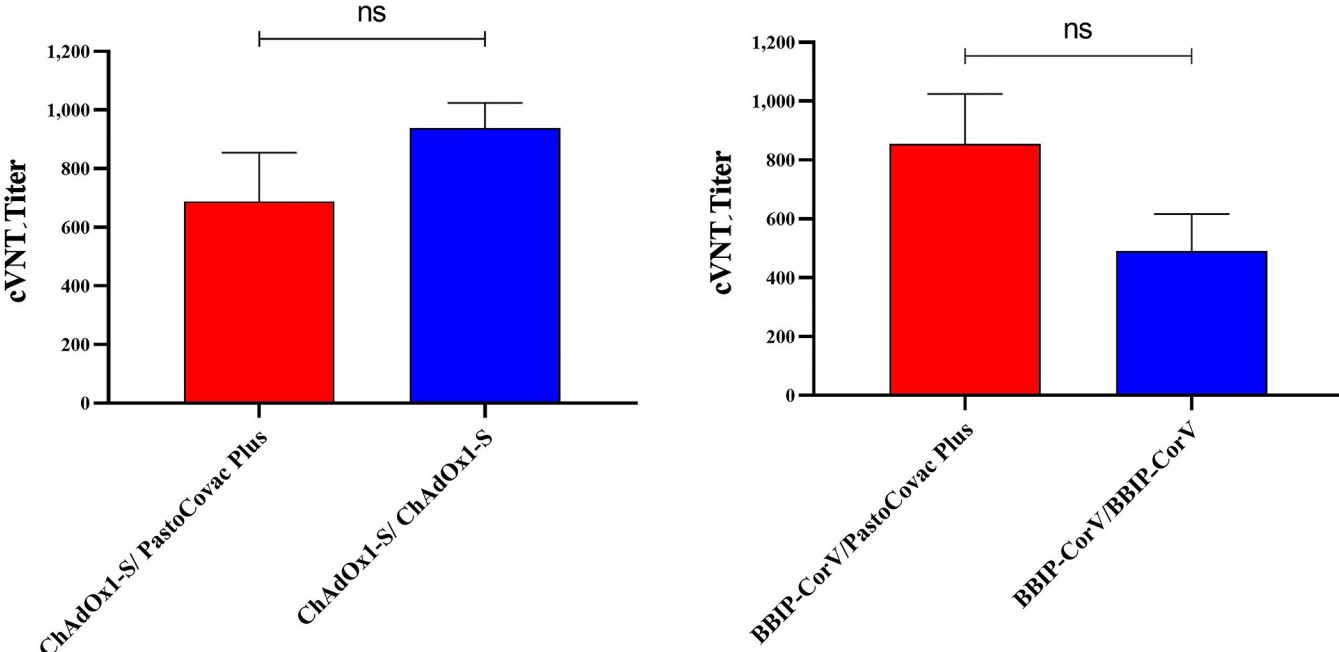

**Fig 4. Comparison of virus neutralizing test between the applied boosters.** Sera from vaccinated individuals with complete schedule were evaluated (cVNT50: GMT, 95% CI) against Omicron BA.5 variant. Ns indicates no significant between heterologous or homologous ChAdOx1-S (*P*: 0.42) nor heterologous/homologous BBIP-CorV (*P*: 0.08).

39.7% of the individuals had at least one adverse event which were mostly among the homologous ChAdOx1-S group. In fact, the higher incidence of fatigue, myalgia and chills were recorded among homologous ChAdOx1-S comparing to the heterologous ChAdOx1-S /PastoCovac Plus group (S3 Table).

## Discussion

A heterologous prime-boost strategy is applied as an effective approach to enhance immune responses against a broad variety of infectious diseases like human immunodeficiency virus (HIV) and hepatitis B virus (HBV), therefore, it is not a newly launched method [21,24]. The primary ideas of heterologous COVID-19 vaccine strategy stemmed from the reported side-effects of ChAdOx1-S and Ad26.COV2.S (Janssen) vaccines [25,26].

Recent studies have shown that heterologous prime-boost vaccination may provide a better protective effect against COVID-19 infection [27–29], while identifying the most effective booster vaccine has remained unclear, yet. Data on the impact of a heterologous protein sub-unit booster on antibodies response in volunteers primarily vaccinated with ChAdOx1-S or BBIBP-CorV vaccines are limited.

In this study, we performed a comparative analysis of the safety and immunogenicity of PastoCovac Plus as the booster dose in previously vaccinated people with ChAdOx1-S and BBIBP-CorV in parallel with the homologous booster shots. All the applied boosters induced specific antibody levels in ChAdOx1-S or BBIBP-CorV-primed volunteers; however, the immunogenicity of the boosters varied. The heterologous boosting regimens in which a protein subunit vaccine was applied as the booster dose in primed individuals with Adeno virus based vaccine, ChadOX1-S, or inactivated virus-based one, BBIP-CorV, led to greater anti-Spike IgG mean rise compared to the homologous ones. Furthermore, anti-Spike IgG and neutralizing antibodies fourfold rise were seen in 92% and 56% of primed individuals with BBIBP-CorV, respectively after PastoCovac Plus injection supporting mix-and-match regimen.

In agreement with the present findings, our previous research also showed that PastoCovac Plus application significantly enhanced the levels of anti-Spike and neutralizing antibodies in primarily vaccinated healthcare workers with COVAXIN vaccine [30]. Moreover, a recent study demonstrated that Novavax (as a protein subunit vaccine) induced a higher mean rise of anti-spike IgG among ChAdOx1 nCoV-19-primed participants as heterologous regimen compared to the homologous one [31].

SOBERANA 02 (PastoCovac) assessment in showed that it is safe and immunogenic either through homologous scheme in which individuals got three doses of the same vaccine or got a booster dose of SOBERANA Plus (PastoCovac Plus). Nevertheless, the highest immune responses were detected in the heterologous protocol [22,32]. Moreover, they demonstrated that neutralizing capacity against D614G, the circulating variant of the study time which was significantly higher after the booster does. cVNT test in our data indicated that PastoCovac Plus has the potency to neutralize Omicron BA.5 variant. In addition, the multicenter, randomized phase 3 trial in 6 cities of Iran showed acceptable PastoCovac vaccine efficacy against symptomatic form of COVID-19 infection and its related severe infections [33].

Apart from the immunogenicity assessment, the adverse events of the booster doses must be taken to attention [34–36]. In addition to clinical trials and primary safety reports of COIVD-19 vaccines, there have been a growing number of case reports experiencing adverse events post-vaccination [37].

The study by Jin P et al, was carried out to evaluate the safety and immunogenicity of heterologous immunization with a recombinant adenovirus type-5-vectored COVID-19 vaccine

(Convidecia) and (ZF2001) a protein-subunit-based vaccine in healthy adults. The reported adverse events were all mild, and similar to our findings, the most common adverse local reaction was pain in the injection site. Heterologous application of ZF2001 in primed individuals with Convidecia could induce robust immune responses [38].

In the present research, the most common local and systemic adverse events were local pain and fatigue/weakness, respectively. Comparing the study arms, the occurrence of fatigue/weakness, myalgia and chills were significantly higher in ChAdOx1-S homologues prime-boost recipients compared to PastoCovac Plus heterologous group which highlights the well safety of this booster shot.

There was no meaningful difference of adverse events rate between BBIBP-CorV primed groups and the parallel heterologous one which represents the equal safety of PastoCovac Plus and BBIBP-CorV as booster vaccines. In addition, a long-term follow-up study on the investigated individuals in our studies presented that PastoCovac Plus had no serious unsolicited adverse events 6 months post-injection [39].

Therefore, PastoCovac Plus, as a protein subunit vaccine with lower rate of side effects, appears to be better option as a booster dose, particularly after viral vectored-vaccine injection.

Apart from the significant results of the study, there were some limitations as well. It was not an interventional study and individuals were free to choose the booster type and this issue resulted in a low-populated homologous group in primed individuals with ChAdOx1-S though we considered the same sample size in each study arm. This also resulted in differences in baseline antibody titers. We adjusted for the baseline values to control for the difference in baseline values. Besides, the results would be ideally evaluated for other platforms like mRNA-based vaccine.

We also recommend long-term follow-up studies to evaluate the vaccine safety and the rate of re-infections after vaccination and its possible effect on immune responses among different COVID-19 vaccinated populations.

## Conclusions

In conclusion, our findings demonstrated that boosting PastoCovac Plus application on previously primed BBIBP-CorV individuals is highly immunogenic, well tolerated and without any serious adverse events. Furthermore, the results indicated that the immunogenicity of Pasto-Covac Plus is equivalent to ChAdOx1-S with a high rate of anti-Spike IgG and neutralizing antibody induction and lower rate of adverse events. According to the recent data and new SARS-CoV-2 variants which make the booster doses as highly recommended agents, protein subunit vaccines could bring promising immunity via heterologous vaccination.

## Supporting information

**S1 Datasheet. Anti-Spike and Neutralization Abs before and after the booster shots in vaccinated individuals.**
(XLSX)

**S1 Table. Association of age and history of COVID-19 infection with mean rise and four-fold rise rate of specific antibodies between the ChAdOx1-S primed groups.**
(DOCX)

**S2 Table. Association of age and history of COVID-19 infection with mean rise and sero-conversion rate of anti-spike IgG antibodies between the BBIBP-CorV -primed groups.**
(DOCX)

**S3 Table. Adverse events data of following the homologous and heterologous booster vaccination.**
(DOCX)

## Acknowledgments

We would like to thank Dr. Saeideh Haji Maghdoodi for her advice on statistical analysis. We also appreciate the participants who dedicated their time and effort to this study.

## Author Contributions

**Conceptualization:** Alireza Biglari, Amitis Ramezani.

**Data curation:** Sana Eybpoosh, Fatemeh Ashrafian, Mona Sadat Larijani, Mohammad Banifazl, Ehsan Mostafavi, Anahita Bavand.

**Formal analysis:** Sana Eybpoosh.

**Funding acquisition:** Amitis Ramezani.

**Methodology:** Sana Eybpoosh, Sarah Dahmardeh, Ehsan Mostafavi, Anahita Bavand.

**Project administration:** Alireza Biglari, Rahim Sorouri, Amitis Ramezani.

**Resources:** Rahim Sorouri.

**Supervision:** Mostafa Salehi-Vaziri, Delaram Doroud, Amitis Ramezani.

**Validation:** Vicente Verez-Bencomo, Maria Eugenia Toledo-Romani, Carmen Valenzuela Silva, Delaram Doroud.

**Writing – original draft:** Fatemeh Ashrafian, Mona Sadat Larijani.

**Writing – review & editing:** Mona Sadat Larijani, Mostafa Salehi-Vaziri, Ehsan Mostafavi, Amitis Ramezani.

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
