## [Decision Letter · Decision Letter 0]

12 Apr 2023

Dear Prof. Ramezani,

Thank you very much for submitting your manuscript "Immunogenicity and safety of heterologous boost immunization with PastoCovac Plus® against COVID-19 in ChAdOx1-S or BBIBP-CorV primed individuals" for consideration at PLOS Pathogens. As with all papers reviewed by the journal, your manuscript was reviewed by members of the editorial board and by several independent reviewers. In light of the reviews (below this email), we would like to invite the resubmission of a significantly-revised version that takes into account the reviewers' comments.

The authors believed there is some value in comparing between different vaccine platforms. However, they raised several major concerns. In particular, the RBD ELISA cannot be the sole method of assessing the platforms, a virus neutralization assay should be included. Please also address the other concerns noted. 

We cannot make any decision about publication until we have seen the revised manuscript and your response to the reviewers' comments. Your revised manuscript is also likely to be sent to reviewers for further evaluation.

Sincerely,

Benhur Lee

Section Editor

PLOS Pathogens

Benhur Lee

Section Editor

PLOS Pathogens

Kasturi Haldar

Editor-in-Chief

PLOS Pathogens

orcid.org/0000-0001-5065-158X

Michael Malim

Editor-in-Chief

PLOS Pathogens

orcid.org/0000-0002-7699-2064

Reviewer's Responses to Questions

**Part I - Summary**

Reviewer #1: In this manuscript the authors compare raise in virus neutralizing and binding antibody titers in serum obtained from study participants that had received ChAdOx1-S (AstraZeneca) or BBIBP-CorV (SinoPharm) before followed by a homologous booster vaccination or heterologous vaccination booster with the recombinant PastoCovac vaccine. The outcome of such studies is informative and important since heterologous prime/boost regimes are currently being rolled out as it became clear that not all COVID-19 vaccines that were initially used early during the pandemic had similar vaccine efficacy, and this might be corrected using a heterologous booster strategy. The manuscript is written well however I have some major concerns.

Reviewer #2: Eybpoosh et al report on a study measuring the immunogenicity and safety of a recombinant protein based COVID-19 vaccine (PastoCovac Plus) that was administered as a booster in individuals that had received two doses of ChAdOx1-S or BBIBP-CorV previously. Two additional groups of individuals received a third dose of ChAdOx1-S or BBIBP-CorV. Total Spike specific antibody responses and neutralizing antibody titers were measured. No differences in antibody titers were observed between all four groups, although the highest titer was detected in the heterologous boost groups. The highest fold increase was observed in the BBIBP-CorV + PastoCovac Plus, but this effect is likely caused by the lower antibody response in these individual prior to the boost. Overall, the abstract and the title of the manuscript are supported by the data.

Major comments:

• Please add P-values to the figures to indicate significance (or lack thereof) within and between groups.

• Some individuals were infected during the course of the study and it is not clear if these were excluded or included in the final analysis.

• The neutralization assay is unknown to me and the data seem odd. Is there an upper limit to this assay? What kind of assay is it, a pseudovirus neutralization assay? Is it possible to provide the serum dilutions that resulted in 50% or 100% virus neutralization? The data is very tight at the top and I wonder if you are losing resolution.

• The recombinant protein vaccine can be described a bit more in the manuscript. It is composed of the RBD alone, which can affect the data in Figure 2 where you are measuring the antibody response to the entire Spike protein.

• Line 264 you mention a that "PastoCovac Plus leads to a significant anti-spike IgG mean rise compare to the homologous booster recipients". What is this statement based on? Based on Figure 2, I find that hard to be true for the ChAdOx1-S groups.

**Part II – Major Issues: Key Experiments Required for Acceptance**

Reviewer #1: The different vaccine types that study participants have received need to be better described. Along the same line, the results should be discussed in light of the different vaccine platforms that were used (adenovirus, recombinant protein, inactivated virus vaccine + adjuvant).

The potential impact of COVID-19 history is not clearly represented. Which samples in figures 2 and 3 are from individuals with previous COVID-19 exposure. A color coding system might help here.

Why are the antibody titers (both ELISA and neutralization) before boost so different between the homologous and heterologous boosted groups for the BBIBP-CoV groups? The differences between these pre-booster titers are even significant according to Table 2. The fold rise in titers in boost is the heterologous boost group is large but the initial pre-boost titers are very low as well.

Sup. Table 1.: COVID-19 History for the homologous ChadOx1-S: n=13 for Yes and 14 for No. These numbers are the opposite in the other parts of the column for the same samples. Are these numbers swapped by accident?

Since there seems to be COVID-19 history available during and after the study time points, the authors should elaborate more on vaccine efficacy/correlation with observed antibody titers and protection from infection.

Reviewer #2: An ELISA against the RBD alone could be useful and re-analyzing or re-doing the virus neutralization assay

**Part III – Minor Issues: Editorial and Data Presentation Modifications**

Reviewer #1: none

Reviewer #2: (No Response)

PLOS authors have the option to publish the peer review history of their article (what does this mean?). If published, this will include your full peer review and any attached files.

Reviewer #1: No

Reviewer #2: No
---

## [Decision Letter · Decision Letter 1]

10 Oct 2023

Dear Prof. Ramezani,

We are pleased to inform you that your manuscript 'Immunogenicity and safety of heterologous boost immunization with PastoCovac Plus against COVID-19 in ChAdOx1-S or BBIBP-CorV primed individuals' has been provisionally accepted for publication in PLOS Pathogens.

Best regards,

Benhur Lee

Section Editor

PLOS Pathogens

Benhur Lee

Section Editor

PLOS Pathogens

Kasturi Haldar

Editor-in-Chief

PLOS Pathogens

orcid.org/0000-0001-5065-158X

Michael Malim

Editor-in-Chief

PLOS Pathogens

orcid.org/0000-0002-7699-2064

Reviewer Comments (if any, and for reference):

Reviewer's Responses to Questions

**Part I - Summary**

Reviewer #1: my previous concerns were addressed adequately by the authors

**Part II – Major Issues: Key Experiments Required for Acceptance**

Reviewer #1: NA

**Part III – Minor Issues: Editorial and Data Presentation Modifications**

Reviewer #1: NA

PLOS authors have the option to publish the peer review history of their article (what does this mean?). If published, this will include your full peer review and any attached files.

Reviewer #1: **Yes: **Michael Schotsaert

---

## [Editor Report · Acceptance letter]

18 Oct 2023

Dear Prof. Ramezani,

We are delighted to inform you that your manuscript, "Immunogenicity and safety of heterologous boost immunization with PastoCovac Plus against COVID-19 in ChAdOx1-S or BBIBP-CorV primed individuals," has been formally accepted for publication in PLOS Pathogens.

Best regards,

Kasturi Haldar

Editor-in-Chief

PLOS Pathogens

orcid.org/0000-0001-5065-158X

Michael Malim

Editor-in-Chief

PLOS Pathogens

orcid.org/0000-0002-7699-2064